# Deploying an *In Vitro* Gut Model to Assay the Impact of the Mannan-Oligosaccharide Prebiotic Bio-Mos on the Atlantic Salmon (*Salmo salar*) Gut Microbiome

Raminta Kazlauskaite,[a] Bachar Cheaib,[a] Joseph Humble,[a] Chloe Heys,[a] Umer Zeeshan Ijaz,[b] Stephanie Connelly,[b] William T. Sloan,[b] Julie Russell,[b] Laura Martinez-Rubio,[c] John Sweetman,[d] Alex Kitts,[a] Philip McGinnity,[e,f] Philip Lyons,[d] Martin S. Llewellyn[a]

[a]Institute of Behaviour, Animal Health and Comparative Medicine, Graham Kerr Building, University of Glasgow, Glasgow, Scotland
[b]School of Engineering, University of Glasgow, Glasgow, Scotland
[c]Mowi ASA, Bergen, Norway
[d]Alltech Inc., Lexington, Kentucky, USA
[e]School of Biological, Earth and Environmental Sciences, University College Cork, Cork, Ireland
[f]Marine Institute, Foras na Mara, Newport, Ireland

**ABSTRACT** Alpha mannose-oligosaccharide (MOS) prebiotics are widely deployed in animal agriculture as immunomodulators as well as to enhance growth and gut health. Their mode of action is thought to be mediated through their impact on host microbial communities and their associated metabolism. Bio-Mos is a commercially available prebiotic currently used in the agri-feed industry, but studies show contrasting results of its effect on fish performance and feed efficiency. Thus, detailed studies are needed to investigate the effect of MOS supplements on the fish microbiome to enhance our understanding of the link between MOS and gut health. To assess Bio-Mos for potential use as a prebiotic growth promoter in salmonid aquaculture, we have modified an established Atlantic salmon *in vitro* gut model, SalmoSim, to evaluate its impact on the host microbial communities. The microbial communities obtained from ceca compartments from four adult farmed salmon were inoculated in biological triplicate reactors in SalmoSim. Prebiotic treatment was supplemented for 20 days, followed by a 6-day washout period. Inclusion of Bio-Mos in the media resulted in a significant increase in formate ($P = 0.001$), propionate ($P = 0.037$) and 3-methyl butanoic acid ($P = 0.024$) levels, correlated with increased abundances of several, principally, anaerobic microbial genera (*Fusobacterium, Agarivorans, Pseudoalteromonas*). DNA metabarcoding with the 16S rDNA marker confirmed a significant shift in microbial community composition in response to Bio-Mos supplementation with observed increase in lactic acid producing *Carnobacterium*. In conjunction with previous *in vivo* studies linking enhanced volatile fatty acid production alongside MOS supplementation to host growth and performance, our data suggest that Bio-Mos may be of value in salmonid production. Furthermore, our data highlights the potential role of *in vitro* gut models to complement *in vivo* trials of microbiome modulators.

**IMPORTANCE** In this paper we report the results of the impact of a prebiotic (alpha-MOS supplementation) on microbial communities, using an *in vitro* simulator of the gut microbial environment of the Atlantic salmon. Our data suggest that Bio-Mos may be of value in salmonid production as it enhances volatile fatty acid production by the microbiota from salmon pyloric ceca and correlates with a significant shift in microbial community composition with observed increase in lactic acid producing *Carnobacterium*. In conjunction with previous *in vivo* studies linking enhanced volatile fatty acid production alongside MOS supplementation to host growth and performance, our data suggest that Bio-Mos may be of value in salmonid production. Furthermore, our data highlights the potential role of *in vitro* gut models to augment *in vivo* trials of microbiome modulators.

**KEYWORDS** MOS, Atlantic salmon, microbiome, *in vitro*, gut model, NGS

**Ad Hoc Peer Reviewer** Emma Hernandez Sanabria, VIB

Address correspondence to Raminta Kazlauskaite, r.kazlauskaite.1@research.gla.ac.uk.

The authors declare no conflict of interest.

Since the late 1970s, the salmon aquaculture sector has grown significantly, currently exceeding 1 million tonnes of salmon produced per year (1). In aquaculture environments, particularly were fish are reared in sea cages, fish are exposed to abiotic conditions and biotic interactions that are extensively different from the wild, such as changes in temperature and salinity, and close contact between animals that can favor potential disease outbreaks (2), as well as chronic stress through physical aggression and overcrowding (3, 4). The rapid expansion of the aquaculture sector requires means to promote efficient feed conversion, reduce the need for medical treatments and reduce waste discharges while also improving farmed fish quality.

In order to mitigate disease outbreaks and improve feed conversion, prebiotics are widely deployed in agriculture and aquaculture settings (5 to 7). Prebiotics are defined as nondigestible food additives that have a beneficial effect on the host by stimulating growth and activity of bacterial communities within the gut that improve animal health (8). One prebiotic type used in aquaculture is alpha-mannooligosaccharides (MOS); these glycans are made of glucomannoprotein-complexes derived from the outer layer of yeast cell walls (*Saccharomyces cerevisiae*) (9). MOS compounds were shown to improve gut function and health by increasing villi height, evenness and integrity in chickens (10, 11), cattle (12) and fish (13). MOS supplementation in monogastrics has been reported to drive changes in host-associated microbial communities and increase performance (8 to 9%) in rainbow trout (14–16). Associated increase of volatile fatty acids (VFA), that can have beneficial knock-on effects in terms of host metabolism and gut health, has been reported (17).

There are limited number of studies investigating the effect of MOS on the fish microbiome (13, 18) with disparities in the observed results that could be partially explained by the duration of MOS supplementation, fish species, age or environmental conditions. For example, it was found that MOS supplemented diets improved growth and/or feed utilization in some studies (19–23), but others found that MOS supplementation did not affect fish performance or feed efficiency (24–26). Detailed studies are needed to investigate the effect of MOS supplements on the fish microbiome to enhance our understanding of the link between MOS and gut health. *In vitro* gut models offer the advantage of doing so in a replicated and controlled environment.

SalmoSim is a salmon gut simulation system that continuously maintains the microbial communities present in the intestine of marine phase Atlantic Salmon (*Salmo salar*) (27). The current study deploys a modified version of SalmoSim designed to evaluate the effect of Bio-Mos (Alltech), a commercially available MOS product, on the microbial communities of the Atlantic salmon small intestine (pyloric cecum) in biological triplicate. The pyloric cecum is the major site of nutrient absorption in the Atlantic Salmon. We investigated microbial composition and fermentation output in the SalmoSim system and showed a significant effect of Bio-Mos supplementation on both.

## RESULTS

In order to explore the impact of the Bio-Mos prebiotic on microbial communities in SalmoSim, microbial amplicons in different experimental phases (Pre-Bio-Mos, Bio-Mos and wash out) were surveyed using HiSeq 2500 amplicon sequencing of the 16S V1 rDNA locus. In total 11.5 million sequence reads were obtained after quality filtering. Alpha diversity metrics (Effective richness in Fig. 1A and effective Shannon diversity in Fig. 1B) indicated that the initial inoculum contained the lowest number of OTUs and had the lowest bacterial richness compared to later sampling time points from SalmoSim system, but these differences were not statistically significant. Furthermore, this figure indicates no statistically significant differences between different experimental phases (Pre-Bio-Mos, Bio-Mos and Wash out) in both terms of effective richness and Shannon diversity. Taken together, diversity and richness estimates suggest nonstatistically significant increase of in the number of detectable microbial taxa as a result of transfer into SalmoSim system, but overwise stable diversity and richness over the different experimental phases.

To provide an overview of microbial composition and variation in the experiment, a PCoA plot was constructed based on Bray-Curtis distanced between samples (Fig. 2A–D).

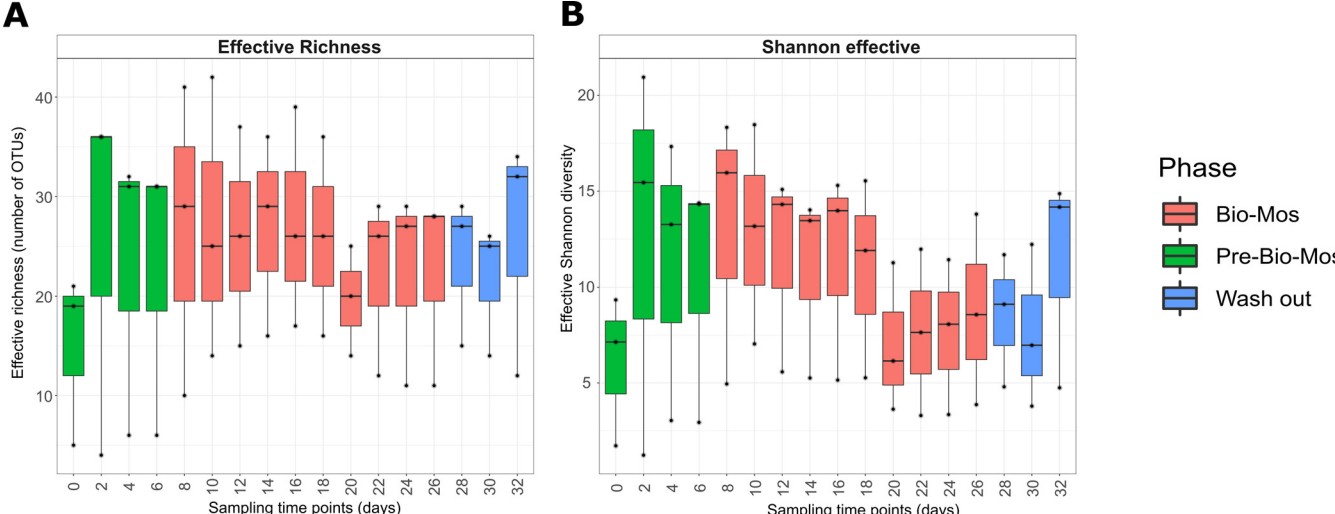

**FIG 1** Alpha-diversity dynamics within the SalmoSim system during exposure to Bio-Mos prebiotics. The figure represents different alpha diversity outputs at different sampling time points (days) from SalmoSim system. Time point 0 represents microbial community composition within initial SalmoSim inoculum from the pregrown stable bacterial communities, time points 2–6 identifies samples from SalmoSim system fed on Fish meal diet alone (Pre-Bio-Mos: green), time points 8–26 identifies samples from SalmoSim system fed on Fish meal diet with addition of Bio-Mos (Bio-Mos: red), and time points 28–32 identifies samples from wash out period while SalmoSim was fed on feed without addition of prebiotic (Wash out: blue). A: visually represents effective richness (number of OTUs), B: represents effective Shannon diversity.

Biological replicate (the founding inoculum of each SalmoSim run) appears to be a major driver of community composition in the experiment (Fig. 2A). This is supported by Fig. 3 that visually represents various microbial composition within different fish. Only when individual SalmoSim replicates were visualized separately in PCoA plots, do the changes to microbial communities in response to the different experimental phases become apparent (Fig. 2B–D). These results indicate that bacterial communities shift from Pre-Bio-Mos to Bio-Mos, but they remain fairly stable (statistically similar, $P > 0.05$ in majority of cases) between Bio-Mos and Wash out periods as reflected by beta diversity results summarized in Table S1. However, community shifts do not necessarily occur along the same axes in each SalmoSim replicate indicative, perhaps, or a different microbiological basis for that change. This trend is confirmed in Fig. 3 that indicates a more substantial shift in microbial community profile between Pre-Bio-Mos and Bio-Mos phases in Fish 2 and 3, but to the lesser extent in Fish 1. Results were further confirmed by performing beta-diversity analysis using both phylogenetic and ecological distances, both of which indicated statistically significant differences between Pre-Bio-Mos and Bio-Mos phases, but not between Bio-Mos and Wash out periods (Table S1). Furthermore, Table S1 indicates that 149 OTUs were found to be differentially abundant between Pre-Bio-Mos and Bio-Mos phases, while only 5 OTUs were differentially abundant between Bio-Mos and Wash out phases.

To compare experimental phases in more detail, differentially abundant OTUs between various experimental phases were summarized in bar plots at genus level in Fig. 4. The Fig. 4A indicates that between Pre-Bio-Mos and Bio-Mos phases, more OTUs decreased in abundance, rather than increased. The OTUs that differentially increased from Pre-Bio-Mos to Bio-Mos phase were identified to belong to: *Aeromonas* (higher proportion increased [50%] rather than decreased [12.5%]), *Agarivorans*, *Aliivibrio*, *Carnobacterium* (only showed increase and no decrease), *Fusobacterium*, *Pseudoalteromonas*, *Pseudomonas*, *Psychobacter*, and *Shewanella*. Fig. 4B indicates the increase of OTUs belonging to *Enterococcus* and *Thalassospira* genera between Bio-Mos and Wash out, while OTUs belonging to *Micrococcus*, *Myroides* and *Shewanella* genera have decreased.

For the analysis of the microbial community structure throughout the experiment, three OTU co-occurrence networks were analyzed for each phase (Pre-Bio-Mos, Bio-Mos, and Wash out), and the main network characteristics were compared: the degree and centrality betweenness (Fig. S1). Pre-Bio-Mos phase (Fig. S1A) indicates a higher average degree (number of edges per node) than in Bio-Mos or Wash out phases. However, the median of degrees is much higher in Bio-Mos phase compared to Pre-Bio-Mos, suggesting that during

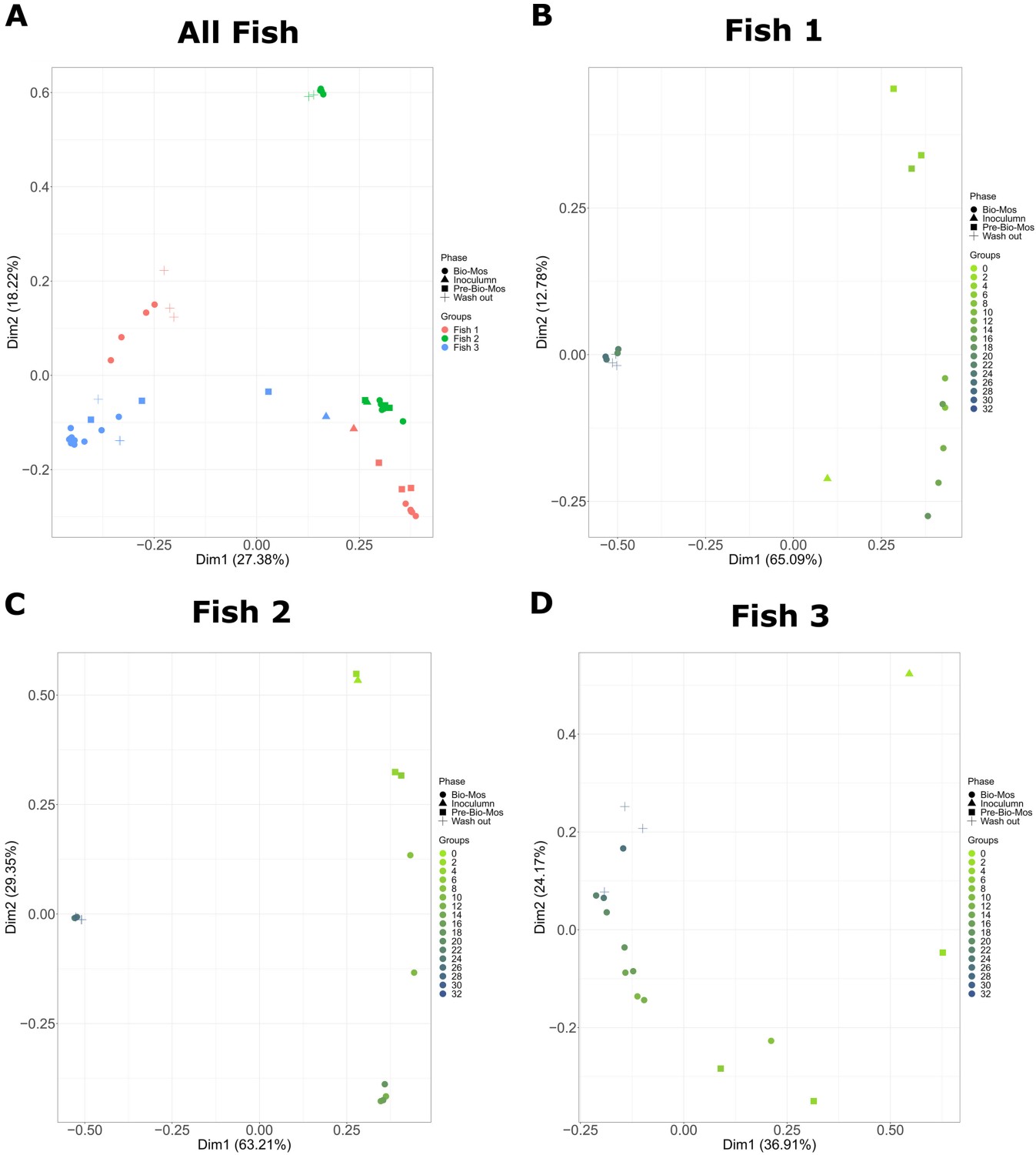

**FIG 2** Beta diversity plots visualizing bacterial communities' dissimilarities within the SalmoSim bioreactors during exposure to Bio-Mos prebiotic. In the PCoA plots, Bray-Curtis distance was used between samples originating from different experimental phases (Inoculum, Pre-Bio-Mos, Bio-Mos and Wash out), annotated with sampling time points and biological replicates. A: represents all sequenced data together for all 3 biological replicates in which different colors represent different biological replicates (samples from pyloric cecum from 3 different fish) and different shapes represent different experimental phases (Inoculum, Pre-Bio-Mos, Bio-Mos and Wash out); B-D: represent sequenced data for each individual biological replicate (B: Fish 1, C: Fish 2, D: Fish 3). In figures B-D: different colors represent different sampling time points and different shapes represent different experimental phases (Inoculum, Pre-Bio-Mos, Bio-Mos and Wash out). Dim 1 is principal coordinate 1 and Dim 2 is principle coordinate 2.

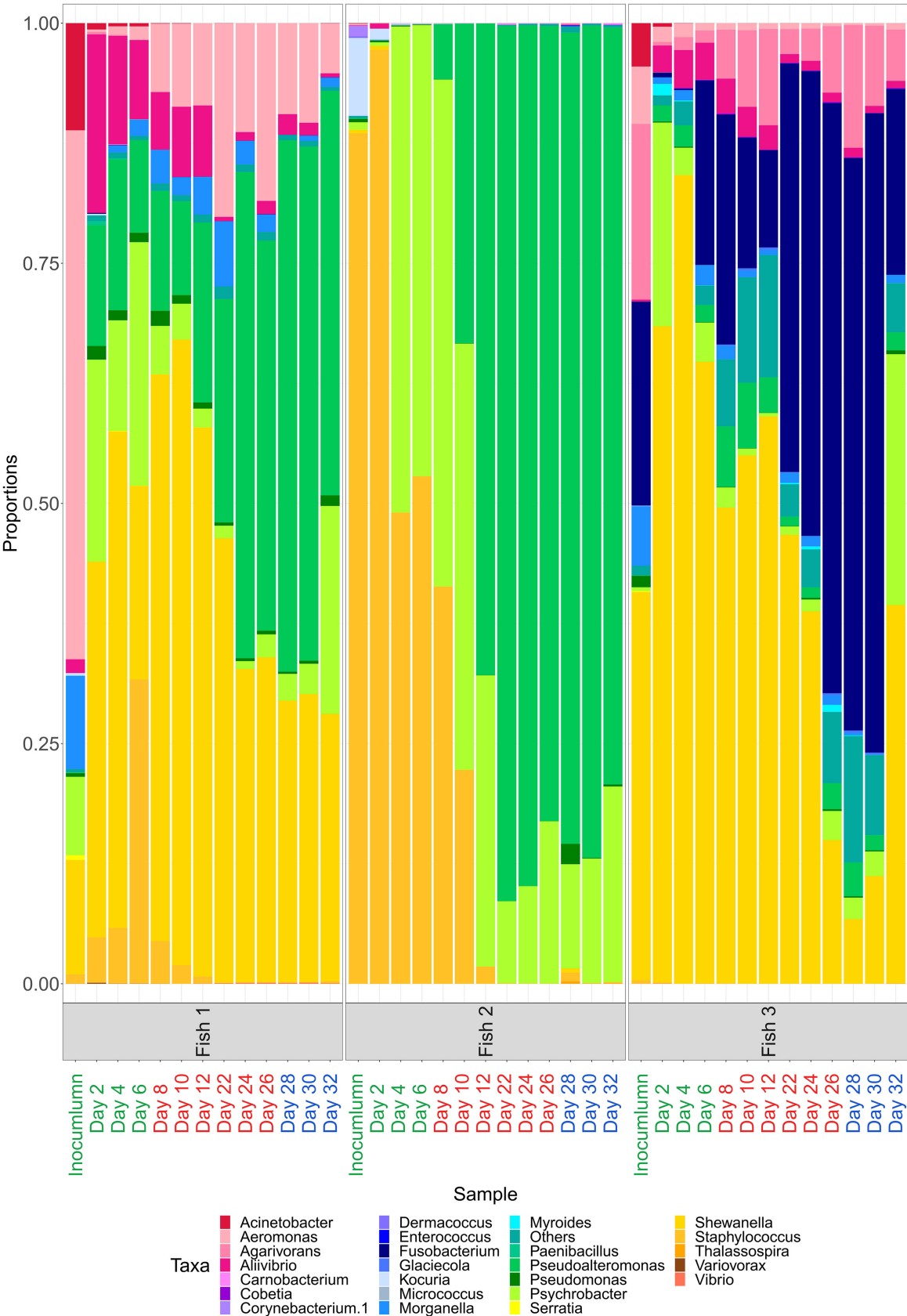

**FIG 3** Microbial composition (25 most common genus + others) among different biological replicates and experimental phases Labels on *x* axis in green represent samples from Pre-Bio-Mos phases, in red samples fed on Bio-Mos phase and in blue samples from Wash out period. Only subset of time points is visualized for each phase: time points 2–6 for Pre-Bio-Mos, 8–12 and 22–24 for Bio-Mos, and 28–32 for Wash out.

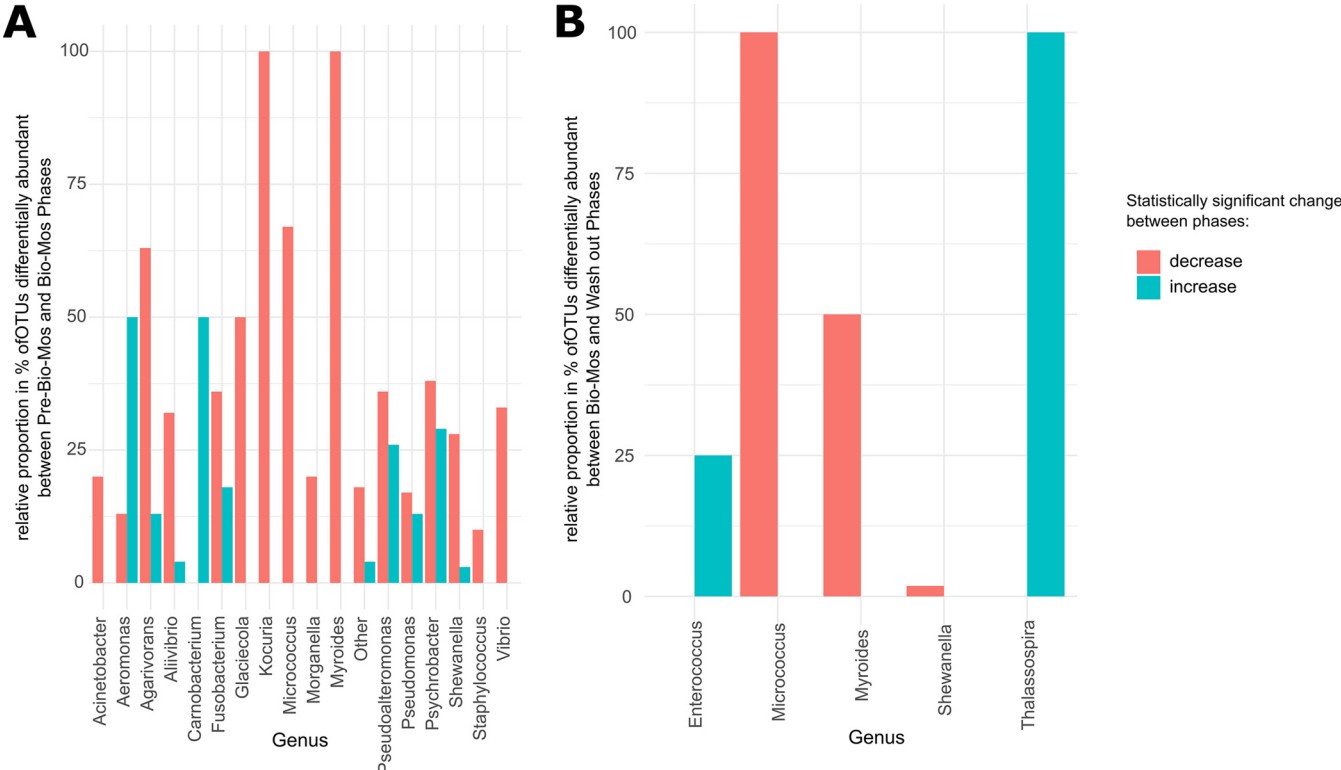

**FIG 4** Differential abundance of OTUs grouped at genus level between different experimental phases (Pre-Bio-Mos, Bio-Mos and Wash out) Differential abundant OTUs grouped at genus level between different experimental phases: Pre-Bio-Mos versus Bio-Mos (A), Bio-Mos versus Wash out (B). Red and blue represents statistically significant ($P < 0.05$) decrease and increase, respectively, between the experimental phases compared.

Pre-Bio-Mos phase there were clusters of interacting OTUs (one cluster with a high degree and another with lower degree). As such, the distribution of connectivity is more uniform in Bio-Mos phases, compared to Pre-Bio-Mos. Moreover, the average of betweenness centralities (centrality measure based on the shortest paths between nodes) are higher in Bio-Mos and Wash out phases compared to Pre-Bio-Mos phase (Fig. S1B).

VFA levels were measured throughout the SalmoSim trial for the stable time points (time points 2, 6 and 8 for Pre-Bio-Mos, time points 22, 24 and 26 for Bio-Mos, and time points 28, 30 and 32 for Wash out period). These results are visually represented in Fig. 5, which indicates that statistically significant increases were found between Pre-Bio-Mos and Bio-Mos phases in formic, propanoic and 3-methylbutanoic acid concentrations. No significant differences in any VFA production by the system was noted between Bio-Mos and Wash out periods.

Bacterial correlates of VFA increases between phases (Fig. 6) were established via Pearson correlation ($r > 0.8$). Results shown in Fig. 7 identify that in the Bio-Mos phase alone, a number of OTUs which showed a strong correlation with various VFAs, had already been picked up by differential abundance analysis (Fig. 4), identifying statistically significant increases. OTUs belonging to *Agarivorans* and *Fusobacterium* genera were found to be positively correlated with propanoic and formic acid, but negatively correlated with 3-methyl butanoic acid. An OTU belonging to *Pseudoalteromonas* genus was found to be positively correlated with propanoic acid, but negatively correlated with 3-methyl butanoic acid, while other OTUs belonging to *Pseudoalteromonas* genus were found to be negatively correlated with propanoic acid. Finally, one OTU belonging to *Fusobacterium* was found to be negatively correlated with 3-methyl butanoic acid. Within Pre-Bio-Mos and Wash out phases, statistically significant Pearson correlations ($r > 0.8$) were also identified between various OTUs and VFAs, however, these OTUs were not also identified as significantly differentially abundant s between different phases of the experiment (Fig. 4, 8B).

Fig. 9 visually summarizes measured ammonia ($NH_3$) concentration changes through the experiment. The data indicate statistically significant increase in ammonia production

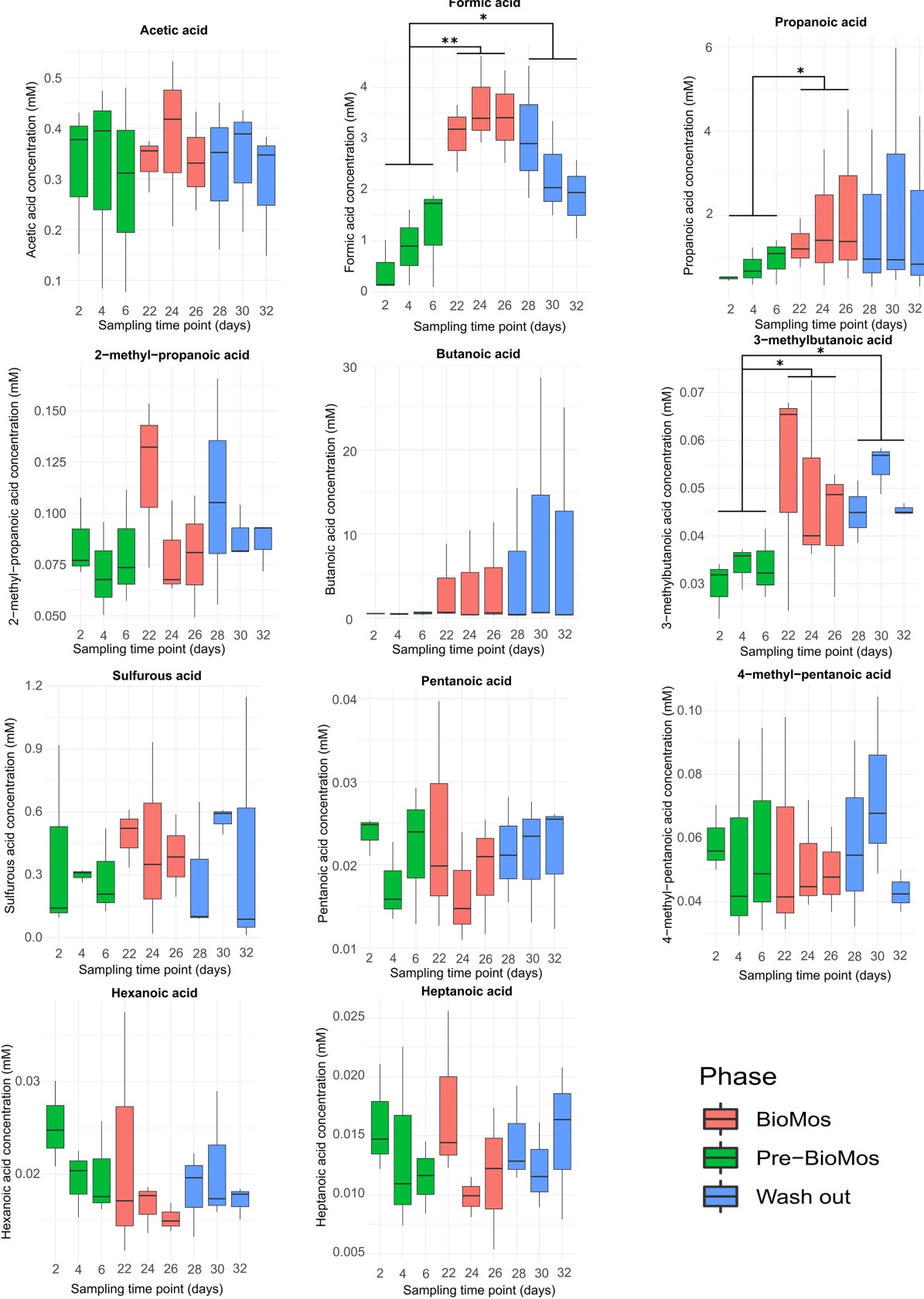

**FIG 5** VFA responses in SalmoSim pyloric cecum compartment after Bio-Mos introduction and subsequent wash out period. The figure above visually represents 11 volatile fatty acid production in three different experimental phases: (i) SalmoSim fed on Fish meal alone

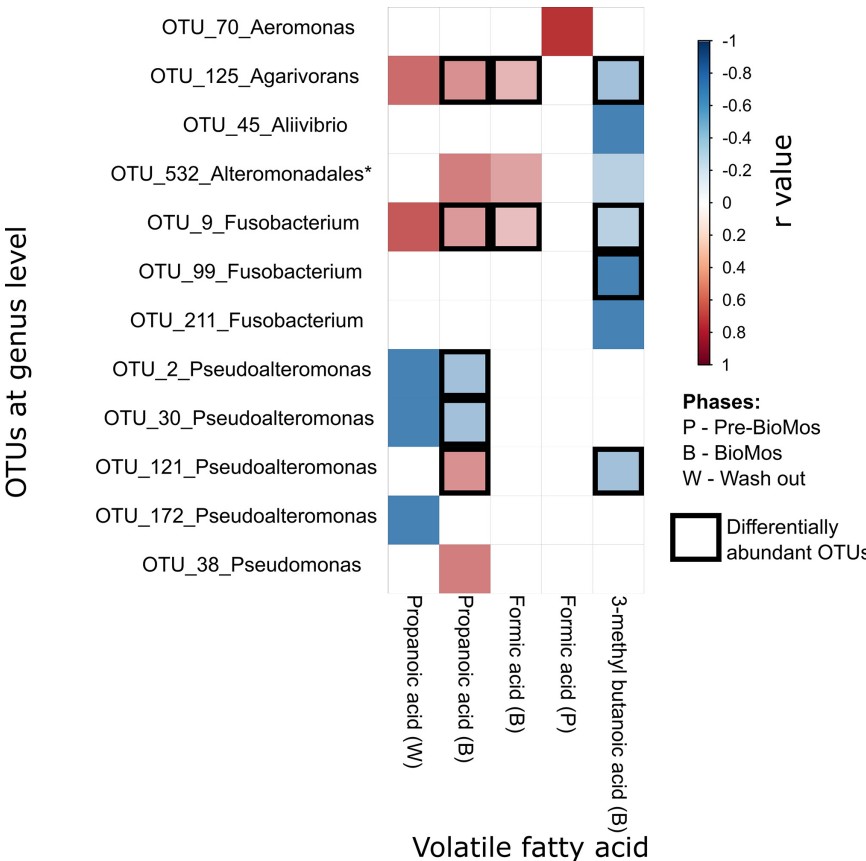

**FIG 6** Pearson correlation coefficients across VFAs and taxonomic variables Statistically significant ($P <$ 0.05) and strongly correlated ($r > 0.8$) Pearson correlation coefficients across a set of VFAs (that showed statistically significant change between feeds: propanoic, formic and 3-methyl butanoic acids) and taxonomic variables (OTUs summarized at genus level apart from * to order level) are shown in various experimental phases (Pre-Bio-Mos, Bio-Mos and Wash out). Blue color represents negative correlation and red color represents positive correlations, respectively. The boxes indicate that these OTUs in differential abundance analysis showed statistically significant increase from Pre-Bio-Mos to Bio-Mos phase.

between time points 2 and 4, and between time points 20 and 22, and statistically significant decrease in ammonia concentration between time points 30 and 32.

## DISCUSSION

Our study aimed at elucidating the effect of a commercially available MOS product (Bio-Mos) on the microbial communities within the gut content of Atlantic salmon using a newly developed artificial salmon gut simulator 'SalmoSim'. The pyloric cecum was chosen on the basis its importance as the main site of nutrient absorption in *S. salar*. Furthermore, previous work has shown little differentiation in microbial communities present in distinct gut compartments (28). Inclusion of Bio-Mos within the tested feed did not affect microbial community diversity and richness in the SalmoSim system, nor did subsequent removal of the prebiotic during wash out. The biological replicate (the founding inoculum of each SalmoSim run) appears to be a major driver of variations in community composition and structure throughout the experiment. This could be partially explained by the fact that feed used in the *in vitro* study was sterile, thus the bacterial communities retrieved within the SalmoSim system

**FIG 5** Legend (Continued)

without prebiotic addition (Pre-Bio-Mos: green), (ii) SalmoSim fed on Fish meal with addition of Bio-Mos (Bio-Mos: red), (iii) wash out period during which SalmoSim was fed on Fish meal without Bio-Mos (Wash out: blue). *x* axis represents the concentration of specific volatile fatty acid (mM) while the *y* axis represents different sampling time points (days). The lines above bar plots represent statistically significant differences between different experimental phases. The asterisks show significance: *, $0.01 \leq P < 0.05$; **, $0.05 \leq P < 0.001$; ***, $P \leq 0.001$.

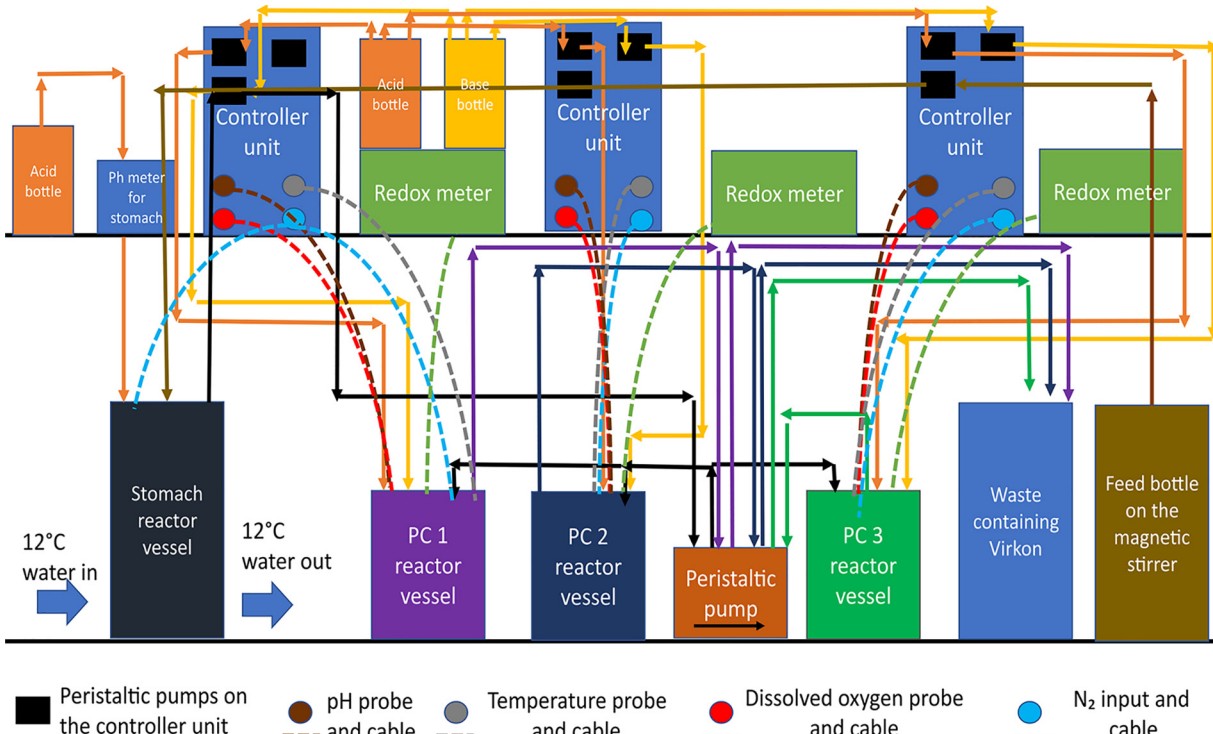

**FIG 7** Artificial gut model system set-up and *in vitro* trial set up. The SalmoSim system designed to run in biological triplicate.

originated only from real salmon inocula as in a previous experiment involving SalmoSim (27). The presence of common feed microbes might have reduced inter-individual variability, but in-feed microbes rarely colonize or establish in the gut (e.g., Heys et al., 2020). Our results indicate that bacterial community composition between Pre-Bio-Mos and Bio-Mos experimental phases was significantly different but was statistically similar between Bio-Mos and Wash out periods. Similar trends were observed in the bacterial activity (VFA production) that showed statistically significant increases in formic, propanoic and 3-methylbutanoic acid concentrations during the shift from Pre-Bio-Mos to Bio-Mos phase, but no statistically significant change in bacterial activity between Bio-Mos and wash-out periods. The lack of change in bacterial composition and activity between Bio-Mos and Wash out period could be explained by the short time frame of the Wash out period, lasting only 6 days, compared to the 20-day Bio-Mos phase. This is potentially not long enough to see a reversal any of changes driven by Bio-Mos. Finally, a statistically significant increase in the ammonia production during Bio-Mos phase was observed at the later time points (between days 20 and 22), followed by the reduction in ammonia concentration during Wash out period (between days 30 and 32), the potential drivers of which we discuss later.

Several studies have shown that in vertebrates (e.g., chicken, mouse, turkey) supplementing feed with MOS increases the production of propionate and butyrate by gut bacteria (29–31), while other studies have not reported any effect of MOS on the VFA production (32). In our study we report a statistically significant increase in the production of formic, propanoic and 3-methylbutanoic acids in the SalmoSim system associated with feed supplemented with Bio-Mos. In humans propionate is commonly absorbed and metabolized by the liver, where it impacts host physiology via regulation of energy metabolism (33). It has also been associated with healthy gut histological development and enhanced growth in fish and shellfish (34, 35). Formic acid, although frequently deployed as a gastric acidifier in monogastrics to limit the growth of enteric

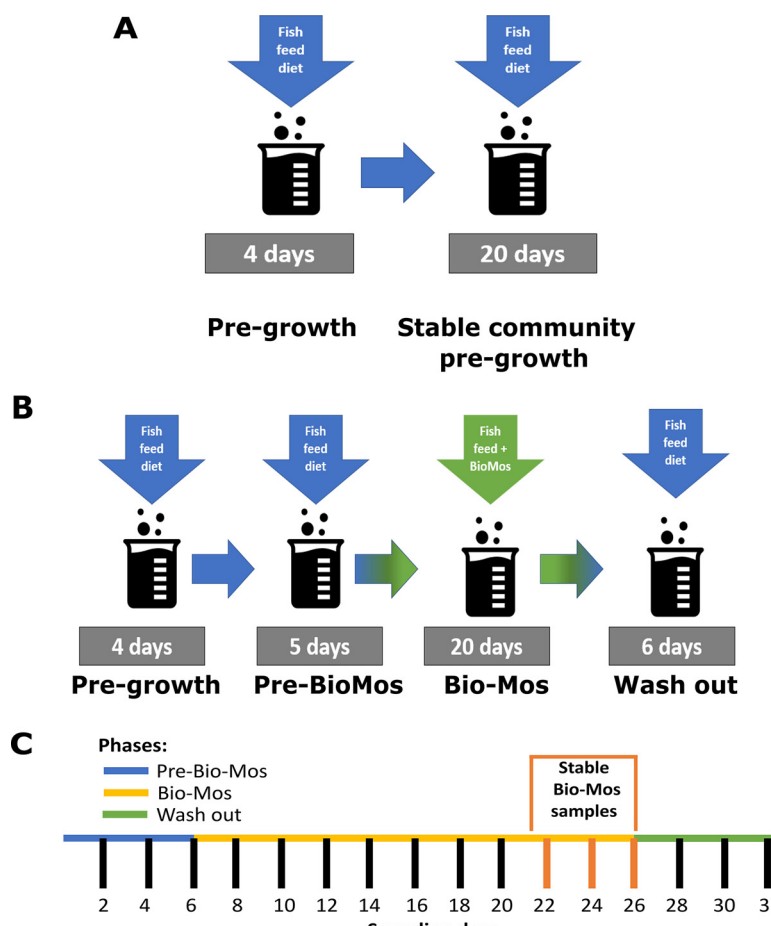

**FIG 8** *In vitro* trial setup. (A) Stable community pregrowth run within the SalmoSim system; (B) main experimental run that involved four stages: (i) pregrowth (without feed transfer for 4 days), (ii) feeding system with Fish meal (Pre-Bio-Mos: 5 days), (iii) feeding system with Fish meal diet supplemented with Bio-Mos (Bio-Mos: 20 days), (iv) wash out period during which system was fed Fish meal without the addition of prebiotic (Wash out: 6 days); (C) SalmoSim sampling time points, which include definition of stable time points for Bio-Mos phase (days 22, 24, and 26 -once bacterial communities had time to adapt to Bio-Mos addition).

pathogens (36), is not known to directly impact host growth or physiology. Similarly, except as the rare genetic disorder that occurs in humans, isovaleric acidemia, where the compound accumulates at high levels in the absence of isovaleric acid-CoA dehydrogenase activity in host tissues (37), isovaleric (3-methylbutanoic) acid is not expected to directly impact host phenotype either.

Further analysis identified that an increase in formic acid during the Bio-Mos phase positively correlated with OTUs belonging to *Agarivorans* (facultative anaerobic) and *Fusobacterium* (anaerobic) genera. While an increase in propanoic acid during the Bio-Mos phase also positively correlated with OTUs belonging to *Agarivorans* (facultative anaerobic) and *Fusobacterium* (anaerobic) genera as well as the *Pseudoalteromonas* (facultative anaerobic) genus, a negative correlation was found with two OTUs belonging to the same *Pseudoalteromonas* (facultative anaerobic) genus. Finally, only negative correlations were identified between the increased amount of 3-methyl butanoic acid in the Bio-Mos phase and OTUs belonging to *Pseudoalteromonas* (facultative anaerobic), *Fusobacterium* (anaerobic) and *Agarivorans* (facultative anaerobic) genera. All of these OTUs were found to not only be correlated with increased VFAs, but also to be differentially abundant between Pre-Bio-Mos and Bio-Mos phases, providing circumstantial evidence for a link between these microbes and the measured metabolites. The causal directionality between these genera and the respective VFAs is hard to

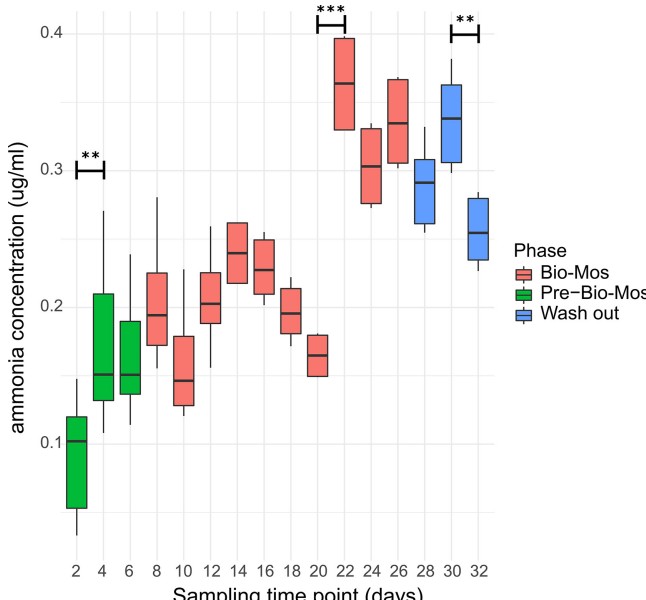

**FIG 9** Ammonia (NH₃) concentration in SalmoSim pyloric cecum compartment throughout experiment. Ammonia (NH₃) production in three different experimental phases: (i) SalmoSim fed on Fish meal alone without prebiotic addition (Pre-Bio-Mos: green), (ii) SalmoSim fed on Fish meal with addition of Bio-Mos (Bio-Mos: red), (iii) wash out period during which SalmoSim was fed on Fish meal without Bio-Mos (Wash out: blue). $x$ axis represents the concentration of ammonia ($\mu$g/mL) while the $y$ axis represents different sampling time points (days). The lines above bar plots represent statistically significant differences between sequential time points. The asterisks show significance: (*, $0.01 \leq P < 0.05$; **, $0.05 \leq P < 0.001$; ***, $P \leq 0.001$).

establish. A strong positive correlation has been found previously in humans between the *Fusobacterium* genus and propanoic acid concentration (38). Propionate is a substrate that can be metabolized by several classes of methanogenic anaerobes (39) and may be driving the growth of the genera noted here. Equally, propionate is a major product of microbial metabolism of amino acids (40), and it is likely here that more efficient protein metabolism in the system by certain genera is driving its abundance. An increase in ammoniacal nitrogen (ammonia) production was noted after the addition of Bio-Mos in all three replicates, albeit with a noticeable lag. Furthermore, although formate, propionate, 3-methyl butanoic acid and ammonia show a downward trend after the removal of Bio-Mos, seemingly a longer wash-out period is required to allow VFA and ammonia to recover their pre-Bio-Mos levels.

Previously published research has suggested that feed supplementation with MOS modulates immune response in animals by stimulation of the production of mannose-binding proteins which are involved in phagocytosis and activation of the complement system (41, 42). Such relationships with host immunity are difficult to predict with a simplified *in vitro* system. It is thought the feed supplementation with MOS elevates the immune response within the host by increasing the lactic acid bacteria (LAB) levels in common carp (43). In the present study, an increase in differential abundance of *Carnobacterium* (LAB bacteria) from Pre-Bio-Mos to Bio-Mos phases was observed. Future work could involve the direct measurement of lactic acid production in the SalmoSim system. This bacterial genus has been proposed as a potential probiotic when present within Atlantic salmon (*Salmo salar*) and rainbow trout (*Oncorhynchus mykiss*) (44). The use of Carnobacteria as probiotics were shown to be correlated with increased survival of the larvae of cod fry and Atlantic salmon fry (45), rainbow trout (46), and salmon (44). A fishmeal-based diet with limited carbohydrate content was used to perform this experiment and has been previously linked to lower abundances of lactic acid producing bacteria compared to microbial gut composition of Atlantic

salmon fed on plant-based feed (47). To enhance LAB growth even further alongside MOS in protein rich diets, some carbohydrate supplementation may be necessary.

Network analysis suggested a change in the distribution of connectivity of the microbial network during the Bio-Mos phase compared to the Pre-Bio-Mos phase. The microbial network during the Bio-Mos phase shows higher modularity (nodes in the network tend to form denser modules), that is also reflected by a higher average of betweenness centralities within the Bio-Mos phase, a measure which represents the degree of interactive connectivity between nodes. Thus, feed supplementation with Bio-Mos may be correlated with more frequent species-species interactions, and a greater stability of network structure within the network. Stable microbial communities are also thought to contribute to pathogen colonization resistance via nutrient niche occupancy (48–50). However, a challenge experiment would be required to test this assertion.

## CONCLUSIONS

Our study indicates the positive correlation between Bio-Mos supplementation and production of propanoic and formic acids, both of which are known to benefit animal microbiome and health (51, 52). Although, our *in vitro* model lacks a host component, previous studies involving the use of gut simulators to analyze the effectiveness of various prebiotics were shown to produce similar results to *in vivo* trials (53, 54). Furthermore, the *in vitro* modulations of VFA production under MOS prebiotic we observe correspond well with previous *in vivo* observations in the literature. Our data, and that of a previous SalmoSim study with parallel *in vivo* and *in vitro* data, highlights the potential usefulness of various *in vitro* gut systems in fin fish aquaculture to study and prescreen the effectiveness of feed additives in advance of, and in complement to *in vivo* trials. As such, models like SalmoSim may be cost-effective for early-stage development of novel feed additives, as well as reducing the number of live fish the are required.

## MATERIALS AND METHODS

**In vivo sample collection and in vitro system inoculation.** Three gut samples from adult starved Atlantic salmon were collected at the same time from the MOWI processing plant in Fort William, Scotland and transferred to the laboratory in an anaerobic box on ice. Samples were placed in an anaerobic hood and 1 g of contents from pyloric ceca compartment were scraped and collected into separate sterile tubes (sample contained both consents and mucosa). Half of each sample was stored in −80°C freezer (as a backup, in case the run needed restarting), while the other half was used as an inoculum for the SalmoSim system. Inocula were prepared for the *in vitro* trial from the inocula pyloric ceca of different individual fish (three biological replicates). Prior to inoculation, inocula were dissolved in 1 mL of autoclaved 35 g/L Instant Ocean Sea Salt solution.

**SalmoSim in vitro system preparation.** The basal *in vitro* system feed medium was prepared as described before (27) by combining the following for a total of 2 liters: 35 g/L of Instant Ocean Sea Salt to generate artificial seawater, 10 g/L of the Fish meal (the same feed as used in [27]), also as in Kazlauskaite et al., 2021, 1 g/L freeze-dried mucus collected and pooled from the pyloric cecum compartments of multiple adult farmed starved Atlantic salmon (different individuals from the ones used in this study as inocula), 2 liters of deionized water. For the Bio-Mos supplemented feed the basal feed medium was supplemented with 0.4% of Bio-Mos (derived from the outer cell wall of *Saccharomyces cerevisiae* strain 1026) dissolved in distilled water. A supplementation level of 0.4% weight by volume was chosen based on previous studies (55, 56). This feed was then autoclave-sterilized, followed by sieving of the bulky flocculate, and finally subjected to a second round of autoclaving. System architecture was prepared as described previously with some modifications (27). In short, appropriate tubes and probes were attached to a two-L double-jacketed bioreactor, and three 500 mL Applikon Mini Bioreactors. Four 1 cm³ aquarium sponge filters were added to each Mini Bioreactor vessel which were then autoclaved, sterilised, and connected as in Fig. 7. Nitrogen gas was periodically bubbled through each vessel to maintain anaerobic conditions. The two-L double jacketed bioreactor and three 500 mL bioreactors were filled with 1.5 liters and 400 mL of feed media, respectively. Once the system was set up, media transfer, gas flow and acid/base addition were undertaken for 20-four hours axenically in order to stabilize the temperature, pH, and oxygen concentration with respect to levels measured from adult salmon. SalmoSim system diagram is visualized in Fig. 7. Physiochemical conditions within the three 500 mL fermenters each inoculated with pyloric cecum materials from different fish were kept similar to the values measured *in vivo* (27): temperature inside the reactor vessels was maintained at 12°C, dissolved oxygen contents were kept at 0% by daily flushing with $N_2$ gas for 20 min, and pH 7.0 by the addition of 0.01 M NaOH and 0.01 M HCl. The 2-L double jacketed bioreactor (representing a sterile stomach compartment) was kept at 12°C and pH at 4.0 by the addition of 0.01 M HCl. During this experiment (apart from the initial pregrowth period), the transfer rate of slurry between reactor vessels was 238 mL per day. Finally, on a daily basis, 1 mL of filtered salmon bile and 0.5 mL of autoclaved 5% mucous solution (5 g of mucous

dissolved in 100 mL of distilled water) were added to the three bioreactors simulating pyloric cecum compartments.

**SalmoSim inoculation and microbial growth.** To generate stable and representative microbial communities for experimentation (27), microbial communities were grown within SalmoSim system during a separate 20-four day run prior to the main experimental run (Fig. 8A). This was achieved by adding fresh inoculum from pyloric ceca to three 500 mL bioreactors which was then pregrown for 4 days without media transfer, followed by 20 days feeding the system at at 238 mL per day feed transfer rate. A volume of 15 mL of the stable communities was collected at the end of this pregrowth period, centrifuged at 3000 $g$ for 10 min and supernatant removed. The pellet was then dissolved in 1 mL of autoclaved 35 g/L Instant Ocean Sea Salt solution, flash frozen in liquid nitrogen for 5 min and stored long term in a −80°C freezer.

**Assaying Bio-Mos impact on microbial communities in the SalmoSim *in vitro* system.** Frozen pregrown stable pyloric ceca samples were thawed on ice and added to the SalmoSim system with each 500 mL bioreactor inoculated using bacterial communities pregrown from a different fish. The system was run in several stages: (i) 4-day initial pregrowth period without feed transfer (Pregrowth), (ii) 5-day period during which SalmoSim was fed without prebiotic (Pre-Bio-Mos), (iii) 20-day period during which SalmoSim was fed on feed supplemented with Bio-Mos (Bio-Mos), (iv) 6-day wash out period during which SalmoSim was fed on Fish meal diet without addition of prebiotic (Wash out). The schematic representation of the experimental design is visually represented in Fig. 8B Sixteen samples were collected throughout the experimental run as described previously (Fig. 8C) (27).

**Genomic DNA extraction and NGS library preparation.** DNA extraction and NGS library preparation protocols were previously described (27, 28). Briefly, the samples collected from SalmoSim system and stable pregrown inocula were thawed on ice and exposed to bead-beating step for 60 s by combining samples with MP Biomedicals 1/4" CERAMIC SPHERE (Thermo Fisher Scientific, USA) and Lysing Matrix A Bulk (MP Biomedicals, USA). Later, DNA was extracted by using the QIAamp DNA Stool kit (Qiagen, Valencia, CA, USA) according to the manufacturer's protocol (57). After, extracted DNA was amplified using primers targeting V1 bacterial rDNA 16s region under the following PCR conditions: 95°C for 10 min, followed by 25 cycles at 95°C for 30 s, 55°C for 30 s and 72°C for 30 s, followed by a final elongation step of 72°C for 10 min. The second-round PCR, which enabled the addition of the external multiplex identifiers (barcodes), involved six cycles only and otherwise had identical reaction conditions to the first round of PCR. This was followed by the PCR product cleanup using Agencourt AMPure XP beads (Beckman Coulter, USA) according to the manufacturers' protocol and gel-purification using the QIAquick Gel Extraction kit (Qiagen, Valencia, CA, USA). Finally, the PCR products were pooled at 10 nM concentration and sent for sequencing using the Novaseq 6000 sequencer. All fastq data have been submitted to the NCBI Short Read Archive PRJNA824256.

**NGS data analysis.** NGS data analysis was undertaken as described previously (27). In short, Sequence analysis was performed with our bioinformatic pipeline as described previously (27), which produced operational taxonomic units (OTUs) table. After, to determine microbial community stability within SalmoSim system over time, two alpha diversity metrics (effective microbial richness and evenness [effective Shannon]) were calculated to analyze using Rhea package (58) and visualized by using microbiomeSeq package based on phyloseq package (59, 60).

To provide an overall visualization of microbial composition across all samples, Principal Coordinates Analysis (PCoA) was performed by using phyloseq package (59, 61) with Bray-Curtis dissimilarity measures calculated by using the vegdist function from the vegan v2.4-2 package (Oksanen et al., 2013). Bray-Curtis distances were calculated for four different data sets: the full data set (containing all biological replicates together), and three different subsets each containing only one of the three biological replicate samples from SalmoSim: Fish inoculum 1, 2, or 3.

To further compare microbial structure between various experimental phases, beta diversity was calculated for two different data sets: (i) all (completed data set containing all the samples sequenced) and (ii) subset (containing all samples for Pre-Bio-Mos and Wash out period, but only stable samplings from Bio-Mos period [time points 22, 12 and 26]). From these data sets ecological distances were computed using Bray-Curtis and Jaccard distances with vegdist() function from the vegan v2.4-2 package (Oksanen et al., 2013). Furthermore, the phylogenetical distances were computed for each data set using GUniFrac() distance (generalized UniFrac) at the 0% (unweighted), 50% (balanced) and 100% (weighted) using the Rhea package (58). Both ecological and phylogenetical distances were then visualized in two dimensions by Multi-Dimensional Scaling (MDS) and nonmetric MDS (NMDS) (62). Finally, a permutational multivariate analysis of variance (PERMANOVA) was performed using distance matrices (including phylogenetic distance) to explain sources of variability in the bacterial community structure as result of changes in recorded parameters (62).

To identify differentially abundant OTUs between various experimental phases (Pre-Bio-Mos, Bio-Mos and Wash out), differential abundance was calculated using microbiomeSeq package based on DESeq2 package (59, 61). Results were then summarized using bar plots at genus level, identifying number of OTUs belonging to specific genus level that increase or decrease between various experimental phases.

To identify OTUs that correlated with measured VFAs, the Pearson correlation coefficient ($r > 0.8$) was calculated between taxonomic variables (OTUs) measured VFA values measured, and visualized using tools supplied by Rhea package within different experimental phases (Pre-Bio-Mos, Bio-Mos, and Wash out) (58).

Finally, in order to analyze microbial community structure within different experimental phases, network analysis using Spearman correlation ($r > 0.8$) was performed (using Cytoscape software) on three data sets: (i) all Pre-Bio-Mos samples, (ii) stable Bio-Mos samples (samples from days 22, 24 and 26), and (iii) all Wash out samples. Key network characteristics were compared between the three experimental phases: i.e., degree and centrality betweenness. All these comparisons were analyzed and visualized using "ggstatsplot" package.

All sequences have been added to the NCBI SRA sequence archive, accession number PRJNA824256.

**Protein fermentation and VFA analysis.** At each sampling point, microbial protein fermentation was assessed by measuring the protein concentration using Thermo Scientific Pierce BCA Protein assay kit (Thermo Fisher Scientific, USA) and the ammonia concentration using Sigma-Aldrich Ammonia assay kit (Sigma-Aldrich, USA). Both methods were performed according to manufacturer protocol by using a Jenway 6305 UV/Visible Spectrophotometer (Jenway, USA). For VFA analysis, nine samples from each pyloric cecum compartment were collected (from 3 biological replicates): 3 samples from Pre-Bio-Mos period (days 2–6), 3 samples from stable time points from the period while SalmoSim was fed on feed supplemented with Bio-Mos (days 22–26), and 3 samples from the Wash out period (days 28–32). VFA sampling was performed as described previously (27). Extracted VFAs were sent for gas chromatographic analysis at the MS-Omics (Denmark).

In order to establish whether VFA concentrations were statistically different between different experimental phases (Pre-Bio-Mos, Bio-Mos and Wash out), a linear mixed effect model was deployed (Model 1) considering time point (sampling time point) and run (biological replicate of SalmoSim system) as random effects.

$$Model\ 1\ =\ lmer(VFA \sim Phase + (1|Time\ point) + (1|Run))$$

Finally, in order to establish whether ammonia production changed throughout experimental run, a linear mixed effect model was deployed (Model 2) treating run biological replicate (of SalmoSim system) as random effect.

$$Model\ 2\ =\ lmer(ammonia\ concentration \sim Time\ point + (1|Run))$$

**Ethics approval and consent to participate.** Animals sampled in the study were euthanised by authorized MOWI employees under Home Officer Schedule 1 of the Animals (Scientific Procedures) Act 1986.

## SUPPLEMENTAL MATERIAL

Supplemental material is available online only.

**SUPPLEMENTAL FILE 1**, PDF file, 1.2 MB.

## ACKNOWLEDGMENTS

We thank Llewellyn Environmental biotechnology laboratories teams for their help in sampling. Big thanks to MOWI team in Fort William Scotland processing plant for letting us collect our samples.

We declare that we have no competing interests.

This research was supported in part by research grants from the BBSRC (grant number BB/P001203/1 & BB/N024028/1), by Science Foundation Ireland, the Marine Institute, and the Department for the Economy, Northern Ireland, under the Investigators Program grant number SFI/15/IA/3028, and by the Scottish Aquaculture Innovation Centre. U.Z.I. is supported by a NERC independent research fellowship (NERC NE/L011956/1) as well as a Lord Kelvin Adam Smith Leadership Fellowship (Glasgow). R.K. is supported by an Alltech PhD Studentship award to the University of Glasgow.

R.K. and M.S.L. conceived the experiment, and R.K., J.H., C.H., J.R., and A.K. performed the *in vitro* experimental procedure and sampling. R.K. performed the DNA extraction and molecular biology experiments, including libraries preparation and quantification. R.K. prepared samples for VFA analysis and analyzed the results. R.K. and B.C. produced and analyzed the NGS results and performed functional diversity analysis. R.K. and M.S.L. wrote the manuscript. All authors reviewed, edited, and approved the final draft of the manuscript.

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
