## [Reviewer comments · Microbiology Spectrum]

Microbiology Spectrum

Deploying an *in vitro* gut model to assay the impact of a mannan-oligosaccharide prebiotic, Bio-Mos® on the Atlantic salmon (*Salmo salar*) gut microbiome.

Raminta Kazlauskaite, Bachar Cheaib, Joseph Humble, Chloe Heys, Umer Ijaz, Stephanie Connelly, William Sloan, Philip McGinnity, Julie Russell, Laura Rubio, John Sweetman, Philip Lyons, Alex Kitts, and Martin Ilewellyn

Corresponding Author(s): Raminta Kazlauskaite, University of Glasgow

Review Timeline:

Submission Date:	January 5, 2022
Editorial Decision:	February 6, 2022
Revision Received:	February 28, 2022
Editorial Decision:	March 11, 2022
Revision Received:	April 7, 2022
Accepted:	April 9, 2022

Editor: Konstantinos Kormas

Reviewer(s): Disclosure of reviewer identity is with reference to reviewer comments included in decision letter(s). The following individuals involved in review of your submission have agreed to reveal their identity: Emma Hernandez Sanabria (Reviewer #3)

Transaction Report:

DOI: <https://doi.org/10.1128/spectrum.01953-21>

February 6, 2022

Dr. Raminta Kazlauskaite
University of Glasgow
Glasgow
United Kingdom

Re: Spectrum01953-21 (Deploying an *in vitro* gut model to assay the impact of a mannan-oligosaccharide prebiotic, Bio-Mos® on the Atlantic salmon (*Salmo salar*) gut microbiome.)

Dear Dr. Raminta Kazlauskaite:

Link Not Available

Sincerely,

Konstantinos Kormas

Journals Department
Reviewer comments:

Reviewer #1 (Comments for the Author):

The authors have addressed all of my queries and I believe that this manuscript is great fit for mSpectrum's readership.

Reviewer #2 (Comments for the Author):

The study describes the impact of MOS administration on salmon microbial communities, using an in-vitro model to understand such effects. The study is presenting an interesting and novel method, recently developed in fish, using in vitro gut simulator, to

evaluate gut microbial composition changes.

The paper is well written, with the results supporting the conclusion, and a thorough discussion.

I have few comments and suggestions for the authors.

1. Starting with the microbial community that the authors targeted, why did they chose to use only the communities from the pyloric caeca? A common strategy when looking at the implications of diet on the gut microbiome and gut health, is to evaluate the distal gut communities. I can understand that the pyloric caeca can be an immunologically active gut site and also potentially highly selective part. So, I wonder if the authors have made the comparison between those different gut parts to see how diet may affect those. The authors in their discussion mention that they found little differentiation between gut compartments, which seems quite surprising given the literature so far.

2. With regard to the diet effect on the gut microbiome changes, I think it would have been quite relevant if the authors would have a control to compare the microbial communities changes with a diet without MOS supplementation running in parallel. In this way, I think the impact of the diet would be more clear and less affected by the impact of history on the microbiome composition. This is something the authors may consider in the future.

3. Line 172: Can the authors briefly report how they build their data? Were there analyzed as OTUs or ASVs?

4. Line 279: How were the co-occurrence networks produced and analysed? Please provides some information.

5. Line 424: Please provided the SRA information of the submitted sequences.

Reviewer #3 (Comments for the Author):

The present study describes the use of an in vitro model to assess the impact of manno-oligosaccharides on the simulated gut environment of the Atlantic salmon. The authors have considered the main abiotic factors that may impact bacterial functionality and they followed the evolution of the community for a long period of time (5 weeks). The advantage of the model is that it seems to remain stable throughout time, but some of its characteristics have not been clarified (i.e., pH range, how the temperature was maintained at 12 {degree sign}C, what was the agitation speed). Many of these variables may have been described elsewhere, but I do believe that they deserve to be mentioned here. Some sentences are lengthy and difficult to follow, and the manuscript will benefit from a revision of the English language grammar and writing style. Manufacturer information about some of the material used is missing and the conclusions are rather general, as the authors just mention briefly that VFA may be related to one or 2 phenotypic characteristics. Lastly, information of the raw data deposition on a database is missing (data accession number?). I think the paper deserves the opportunity to be presented to the audience of Spectrum, after some revisions of the materials and methods section and of the discussion. I have provided the following comments for the authors consideration:

Line 19: "their" associated metabolism.

Line 20: "studies show contrasting results of its effect on fish performance and feed efficiency".

Line 21-25: please reword this sentence or divide into two. In its current form is convoluted and difficult to follow.

Line 26: please reword this sentence, i.e., "microbial communities obtained from caeca compartments from four adult farmed salmon were inoculated in biological triplicate reactors in the SalmoSim. Prebiotic treatment was supplemented for 20 days, followed by a 6-day washout period".

Line 34-35: please briefly mention what kind of results from previous studies have been associated with host growth and performance (e.g., X VFA has been correlated with average daily gain or Y VFA with daily intake). This will make clear for the reader the potential that Bio-Mos has on salmonid production.

Line 37: please replace augment with strengthen. The current sentence reads odd.

Line 42: "the impact of a prebiotic..."

Line 43: "communities, using an in vitro simulator of the gut microbial environment of the Atlantic salmon".

Line 47: what is the relevance of increasing lactic acid production in the gut of salmonids and why Carnobacterium is particularly important? Has lactic acid been correlated with any productive metric? Please add it in this section.

Line 59: "requires means to promote..."

Line 72: Knock-on refers to "carry over" or "side effects"? If this the case, an example of potential associations reported may be useful to be described in this paragraph.

Line 86: please provide information about the manufacturer.

Line 88: "we investigated... and showed that Bio-Mos significantly increases/decreases..."

Line 93: "three gut samples from adult starved Atlantic salmon were collected..."

Line 93: do you have an idea about the amount of sample collected? Did the sample include mucosa and tissue or just the contents?

Line 99: what was the concentration of the inocula? Were they all the same?

Line 101: could you please provide more information about the characteristics of the Sea Salt solution? What was the pH and the osmolality?

Line 107: please standardise the use of "inoculums" or "inocula".

Line 108: one space is missing.
Line 109: please add the manufacturer information of Bio-Mos.
Line 110: distilled?
Line 111: have the authors looked into the impact of 2 rounds of autoclaving on the integrity of the manno-oligosaccharides?
Line 112: what is the composition of the bulky flocculate? Would this subfraction impact differently the bacterial activities on the gut of the salmon? Please provide the rationale for its removal.
Line 127: representing a sterile...
Line 131: was this mucus harvested from different individuals?
Line 138: feeding the system at 238 ml...
Line 142: please confirm that the experiment was composed by 2 stages: a 24-day community generation period and a 35-day test run (4 days of static stabilisation, 5 days of semi-continuous feeding without supplementation, 20 days of feeding with Bio-Mos and 6 days of washout). If this is the case, please avoid using pre-growth period for both the test run and the community generation stage.
Line 278: were the network characteristics significantly different between run step (stabilisation, supplementation, and washout)? How many samples were used for constructing the networks of each run step?
Line 297: correlations?
Line 299: were these correlations significant or the differential abundance analysis detected significantly different abundances? Please reword sentence in lines 298-301.
Line 313: how were these time points selected? Is it because they are at the start of a new phase in the run of the SalmoSim? Please provide a rationale.
Line 351: formate is an intermediate for the formation of propionate. Methanogens and other hydrogenotrophs can also use as [H] donors other intercellular e- carriers such as formate. Please refer to:
<https://www.frontiersin.org/articles/10.3389/fmicb.2020.00589/full>
<https://www.cambridge.org/core/services/aop-cambridge-core/content/view/14246C0DDAD1789FB6DA09B188A83657/S0007114505001492a.pdf/div-class-title-propionate-precursors-and-other-metabolic-intermediates-as-possible-alternative-electron-acceptors-to-methanogenesis-in-ruminal-fermentation-span-class-italic-in-vitro-span-div.pdf>
Line 355: isovalerate can be derived from the fermentation of branched-chain amino acids.

Reviewer #4 (Comments for the Author):

1. As stated by the authors, previous studies showed contrasting results of Bio-Mos affect on fish performance and feed efficiency. The possible reasons may be related to the fish species, concentration of Bio-Mos and intestinal microbiota composition, et al. Different fishes harbor different intestinal microbiota. In vivo study would afford more information concerning the effect of Bio-Mos than in vitro one. The present study used in vitro system to evaluate the influence of Bio-Mos on microbiota composition and metabolites, which can not resolve the inconsistent results of Bio-Mos on fish. Furthermore, based on in vitro results, it is not enough to conclude that Bio-Mos may be of value in salmonid production.
2. Line 179 "vegdist() function", please revise.
3. Authors emphasized that three samples could show the effect of probiotic, but the influence of intestinal microbiota was inconsistent. It is difficult to draw the conclusion how probiotic influenced the microbiota.

Staff Comments:

Preparing Revision Guidelines

Please return the manuscript within 60 days; if you cannot complete the modification within this time period, please contact me. If you do not wish to modify the manuscript and prefer to submit it to another journal, please notify me of your decision immediately so that the manuscript may be formally withdrawn from consideration by Microbiology Spectrum.

Comments for the reviewers

The present study describes the use of an *in vitro* model to assess the impact of manno-oligosaccharides on the simulated gut environment of the Atlantic salmon. The authors have considered the main abiotic factors that may impact bacterial functionality and they followed the evolution of the community for a long period of time (5 weeks). The advantage of the model is that it seems to remain stable throughout time, but some of its characteristics have not been clarified (i.e., pH range, how the temperature was maintained at 12 °C, what was the agitation speed). Many of these variables may have been described elsewhere, but I do believe that they deserve to be mentioned here. Some sentences are lengthy and difficult to follow, and the manuscript will benefit from a revision of the English language grammar and writing style. Manufacturer information about some of the material used is missing and the conclusions are rather general, as the authors just mention briefly that VFA may be related to one or 2 phenotypic characteristics. Lastly, information of the raw data deposition on a database is missing (data accession number?). I think the paper deserves the opportunity to be presented to the audience of Spectrum, after some revisions of the materials and methods section and of the discussion. I have provided the following comments for the authors consideration:

Line 19: “their” associated metabolism.

Line 20: “studies show contrasting results of its effect on fish performance and feed efficiency”.

Line 21-25: please reword this sentence or divide into two. In its current form is convoluted and difficult to follow.

Line 26: please reword this sentence, i.e., “microbial communities obtained from caeca compartments from four adult farmed salmon were inoculated in biological triplicate reactors in the SalmoSim. Prebiotic treatment was supplemented for 20 days, followed by a 6-day washout period”.

Line 34-35: please briefly mention what kind of results from previous studies have been associated with host growth and performance (e.g., X VFA has been correlated with average daily gain or Y VFA with daily intake). This will make clear for the reader the potential that Bio-Mos has on salmonid production.

Line 37: please replace augment with strengthen. The current sentence reads odd.

Line 42: “the impact of a prebiotic...”

Line 43: “communities, using an in vitro simulator of the gut microbial environment of the Atlantic salmon”.

Line 47: what is the relevance of increasing lactic acid production in the gut of salmonids and why *Carnobacterium* is particularly important? Has lactic acid been correlated with any productive metric? Please add it in this section.

Line 59: “requires means to promote...”

Line 72: Knock-on refers to “carry over” or “side effects”? If this the case, an example of potential associations reported may be useful to be described in this paragraph.

Line 86: please provide information about the manufacturer.

Line 88: “we investigated... and showed that Bio-Mos significantly increases/decreases...”

Line 93: “three gut samples from adult starved Atlantic salmon were collected...”

Line 93: do you have an idea about the amount of sample collected? Did the sample include mucosa and tissue or just the contents?

Line 99: what was the concentration of the inocula? Were they all the same?

Line 101: could you please provide more information about the characteristics of the Sea Salt solution? What was the pH and the osmolality?

Line 107: please standardise the use of “inoculums” or “inocula”.

Line 108: one space is missing.

Line 109: please add the manufacturer information of Bio-Mos.

Line 110: distilled?

Line 111: have the authors looked into the impact of 2 rounds of autoclaving on the integrity of the manno-oligosaccharides?

Line 112: what is the composition of the bulky flocculate? Would this subfraction impact differently the bacterial activities on the gut of the salmon? Please provide the rationale for its removal.

Line 127: representing a sterile...

Line131: was this mucus harvested from different individuals?

Line138: feeding the system at 238 ml...

Line 142: please confirm that the experiment was composed by 2 stages: a 24-day community generation period and a 35-day test run (4 days of static stabilisation, 5 days of semi-continuous feeding without supplementation, 20 days of feeding with Bio-Mos and 6 days of washout). If this is the case, please avoid using pre-growth period for both the test run and the community generation stage.

Line 278: were the network characteristics significantly different between run step (stabilisation, supplementation, and washout)? How many samples were used for constructing the networks of each run step?

Line 297: correlations?

Line 299: were these correlations significant or the differential abundance analysis detected significantly different abundances? Please reword sentence in lines 298-301.

Line 313: how were these time points selected? Is it because they are at the start of a new phase in the run of the SalmoSim? Please provide a rationale.

Line 351: formate is an intermediate for the formation of propionate. Methanogens and other hydrogenotrophs can also use as [H] donors other intercellular e⁻ carriers such as formate. Please refer to:

<https://www.frontiersin.org/articles/10.3389/fmicb.2020.00589/full>

<https://www.cambridge.org/core/services/aop-cambridge-core/content/view/14246C0DDAD1789FB6DA09B188A83657/S0007114505001492a.pdf/div-class-title-propionate-precursors-and-other-metabolic-intermediates-as-possible-alternative-electron-acceptors-to-methanogenesis-in-ruminal-fermentation-span-class-italic-in-vitro-span-div.pdf>

Line 355: isovalerate can be derived from the fermentation of branched-chain amino acids.

Dear Dr. Raminta Kazlauskaitė:

<https://spectrum.msubmit.net/cgi-bin/main.plex?el=A7QF7BtSU7A4EFDp2F7A9ftdJMGzl7YeJkdkdwWJhpWKBAZ>

The ASM Journals program strives for constant improvement in our submission and publication process. Please tell us how we can improve your experience by taking this quick Author Survey.

Sincerely,

Konstantinos Kormas

Journals Department
Reviewer comments:

Reviewer #1 (Comments for the Author):

The authors have addressed all of my queries and I believe that this manuscript is great fit for mSpectrum's readership.

Thank you for your comment. We are glad to hear that we addressed all your concerns.

Reviewer #2 (Comments for the Author):

The study describes the impact of MOS administration on salmon microbial communities, using an in-vitro model to understand such effects. The study is presenting an interesting and novel method, recently developed in fish, using in vitro gut simulator, to evaluate gut microbial composition changes.

The paper is well written, with the results supporting the conclusion, and a thorough discussion.

I have few comments and suggestions for the authors.

1. Starting with the microbial community that the authors targeted, why did they chose to use only the communities from the pyloric caeca? A common strategy when looking at the implications of diet on the gut microbiome and gut health, is to evaluate the distal gut communities. I can understand that the pyloric caeca can be an immunologically active gut site and also potentially highly selective part. So, I wonder if the authors have made the comparison between those different gut parts to see how diet may affect those. The authors in their discussion mention that they found little differentiation between gut compartments, which seems quite surprising given the literature so far.

We appreciate the reviewer's comments. As we mention the in text (line 29) we chose the pyloric ceacum as it is the principal site of absorption in the salmon intestine. We also noted in our previous work (<https://pubmed.ncbi.nlm.nih.gov/32033945/>), as well as in (<https://microbiomejournal.biomedcentral.com/articles/10.1186/s40168-021-01134-6>) that microbial communities in the PC and distal gut are very similar to one and other. It is also with noting the BIOMOS has an immunomodulatory function – so the PC is doubly appropriate.

2. With regard to the diet effect on the gut microbiome changes, I think it would have been quite relevant if the authors would have a control to compare the microbial communities changes with a diet without MOS supplementation running in parallel. In this way, I think the impact of the diet would be more clear and less affected by the impact of history on the microbiome composition. This is something the authors may consider in the future.

Thank you for your comment. The wash-in, wash-out expt design provides an internal control. However, we believe a parallel control would also be beneficial and that's definitely one of the improvements that we want to implement in the future.

3. Line 172: Can the authors briefly report how they build their data? Were there analyzed as OTUs or ASVs?

They were analysed as OTUs – this was added to the main manuscript.

4. Line 279: How were the co-occurrence networks produced and analysed? Please provides some information.

Cytoscape software was used – this was added to the main manuscript

5. Line 424: Please provided the SRA information of the submitted sequences.

Should the manuscript be accepted we will upload the sequences. We have added a sentence at the end of the results.

Reviewer #3 (Comments for the Author):

The present study describes the use of an in vitro model to assess the impact of manno-oligosaccharides on the simulated gut environment of the Atlantic salmon. The authors have considered the main abiotic factors that may impact bacterial functionality and they followed the evolution of the community for a long period of time (5 weeks). The advantage of the model is that it seems to remain stable throughout time, but some of its characteristics have not been clarified (i.e., pH range, how the temperature was maintained at 12 {degree sign}C, what was the agitation speed). Many of these variables may have been described elsewhere, but I do believe that they deserve to be mentioned here. Some sentences are lengthy and difficult to follow, and the manuscript will benefit from a revision of the English language grammar and writing style. Manufacturer information about some of the material used is missing and the conclusions are rather general, as the authors just mention briefly that VFA may be related to one or 2 phenotypic characteristics. Lastly, information of the raw data deposition on a database is missing (data accession number?). I think the paper deserves the opportunity to be presented to the audience of Spectrum, after some revisions of the materials and methods section and of the discussion. I have provided the following comments for the authors consideration:

Line 19: "their" associated metabolism.

Updated

Line 20: "studies show contrasting results of its effect on fish performance and feed efficiency".

Updated

Line 21-25: please reword this sentence or divide into two. In its current form is convoluted and difficult to follow.

Updated

Line 26: please reword this sentence, i.e., "microbial communities obtained from caeca compartments from four adult farmed salmon were inoculated in biological triplicate reactors in the SalmoSim. Prebiotic treatment was supplemented for 20 days, followed by a 6-day washout period".

Updated

Line 34-35: please briefly mention what kind of results from previous studies have been associated with host growth and performance (e.g., X VFA has been correlated with average daily gain or Y VFA with daily intake). This will make clear for the reader the potential that Bio-Mos has on salmonid production.

We have added a reference for rainbow trout.

Line 37: please replace augment with strengthen. The current sentence reads odd.

Updated

Line 42: "the impact of a prebiotic..."

Updated

Line 43: "communities, using an in vitro simulator of the gut microbial environment of the Atlantic salmon".

Updated

Line 47: what is the relevance of increasing lactic acid production in the gut of salmonids and why Carnobacterium is particularly important? Has lactic acid been correlated with any productive metric? Please add it in this section.

All this information is included in the references we provide.

Line 59: "requires means to promote..."

Updated

Line 72: Knock-on refers to "carry over" or "side effects"? If this the case, an example of potential associations reported may be useful to be described in this paragraph.

Again – the example is provided in the reference, it is not necessary to provide this information here.

Line 86: please provide information about the manufacturer.

Already mentions above in line 83

Line 88: "we investigated... and showed that Bio-Mos significantly increases/decreases..."

Updated

Line 93: "three gut samples from adult starved Atlantic salmon were collected..."

Updated

Line 93: do you have an idea about the amount of sample collected? Did the sample include mucosa and tissue or just the contents?

Added clarification. In short: 1 g of contents from pyloric caeca compartment were scraped and collected into separate sterile tubes (sample contained both contents and mucosa).

Line 99: what was the concentration of the inocula? Were they all the same?

Yes, 1 g of the content described above dissolved in 1 ml of autoclaved 35 g/L Instant Ocean® Sea Salt solution.

Line 101: could you please provide more information about the characteristics of the Sea Salt solution? What was the pH and the osmolality?

These measurements were not taken, but it was a simple solution preparation by combining 35 g of Instant Ocean® Sea Salt into 1 L of distilled water.

Line 107: please standardise the use of "inoculums" or "inocula".

Thank you for your suggestion, I have standardised them!

Line 108: one space is missing.

Updated

Line 109: please add the manufacturer information of Bio-Mos.

We have provided this information in the introduction.

Line 110: distilled?

Updated

Line 111: have the authors looked into the impact of 2 rounds of autoclaving on the integrity of the manno-oligosaccharides?

Yes – the oligosaccharide is stable – we have confirmed this with the manufacturer.

Line 112: what is the composition of the bulky flocculate? Would this subfraction impact differently the bacterial activities on the gut of the salmon? Please provide the rationale for its removal.

Agreed! It would have been great to include it, but it was blocking the system, thus we had to remove a small proportion of the more bulky flocculates.

Line 127: representing a sterile...

Updated

Line131: was this mucus harvested from different individuals?

Yes, the mucous was harvested from many different individuals, combined and freeze dried as well as autoclaved – we have updated the text

Line138: feeding the system at 238 ml...

Updated

Line 142: please confirm that the experiment was composed by 2 stages: a 24-day community generation period and a 35-day test run (4 days of static stabilisation, 5 days of semi-continuous feeding without supplementation, 20 days of feeding with Bio-Mos and 6 days of washout). If this is the case, please avoid using pre-growth period for both the test run and the community generation stage.

This really is unnecessary – the experimental structure is abundantly clear from Figure 2 and the text. We used a pregrowth period in both stages of the experiment.

Line 278: were the network characteristics significantly different between run step (stabilisation, supplementation, and washout)? How many samples were used for constructing the networks of each run step?

We clearly describe our network methods in the methods section: “Finally, in order to analyse microbial community structure within different experimental phases, network analysis using Spearman correlation ($r > 0.8$) was performed on three datasets: (i) all Pre-Bio-Mos samples, (ii) stable Bio-Mos samples (samples from days 22, 24 and 26), and (iii) all Wash out samples. Key network characteristics were compared between the three experimental phases: i.e., degree and centrality betweenness. All these comparisons were analysed and visualised using “ggstatsplot” package.” The summary of the network characteristics comparison is summarised in Supplementary Figure 7 and described in the results sections.

Line 297: correlations?

Updated

Line 299: were these correlations significant or the differential abundance analysis detected significantly different abundances? Please reword sentence in lines 298-301.

We have made some edits to improve the clarity

Line 313: how were these time points selected? Is it because they are at the start of a new phase in the run of the SalmoSim? Please provide a rationale.

All the time points can be observed in Figure 9 (taken every 2 days have values), but only some of them showed significant differences – which we highlight

Line 351: formate is an intermediate for the formation of propionate. Methanogens and other hydrogenotrophs can also use as [H] donors other intercellular e- carriers such as formate. Please refer to:

<https://www.frontiersin.org/articles/10.3389/fmicb.2020.00589/full>

<https://www.cambridge.org/core/services/aop-cambridge-core/content/view/14246C0DDAD1789FB6DA09B188A83657/S0007114505001492a.pdf/div-class-title-propionate-precursors-and-other-metabolic-intermediates-as-possible-alternative-electron-acceptors-to-methanogenesis-in-ruminal-fermentation-span-class-italic-in-vitro-span-div.pdf>

Many thanks

Line 355: isovalerate can be derived from the fermentation of branched-chain amino acids.

Many thanks

Reviewer #4 (Comments for the Author):

1. As stated by the authors, previous studies showed contrasting results of Bio-Mos affect on fish performance and feed efficiency. The possible reasons may be related to the fish species, concentration of Bio-Mos and intestinal microbiota composition, et al. Different fishes harbor different intestinal microbiota. In vivo study would afford more information concerning the effect of Bio-Mos than in vitro one. The present study used in vitro system to evaluate the influence of Bio-Mos on microbiota composition and metabolites, which can not resolve the inconsistent results of Bio-Mos on fish. Furthermore, based on in vitro results, it is not enough to conclude that Bio-Mos may be of value in salmonid production.

We do not claim that this study acts as a replacement for an in vivo trial, rather as a complement. We have added a sentence at the end of the conclusion to re-affirm this. This whole manuscript describes our validated in vitro model of the salmon gut. Should the reviewer, or other interested parties, wish to read about in vivo studies – there are plenty to choose from in the literature.

2. Line 179 "vegdist() function", please revise.

Updated! Thanks.

3. Authors emphasized that three samples could show the effect of probiotic, but the influence of intestinal microbiota was inconsistent. It is difficult to draw the conclusion how probiotic influenced the microbiota.

We have addressed the issue of inter-individual variation abundantly in the manuscript.

March 11, 2022

Dr. Raminta Kazlauskaite
University of Glasgow
Glasgow G12 8QQ
Glasgow
United Kingdom

Re: Spectrum01953-21R1 (Deploying an *in vitro* gut model to assay the impact of a mannan-oligosaccharide prebiotic, Bio-Mos® on the Atlantic salmon (*Salmo salar*) gut microbiome.)

Dear Dr. Raminta Kazlauskaite:

Link Not Available

Sincerely,

Konstantinos Kormas

Journals Department
Reviewer comments:

Reviewer #2 (Public repository details (Required)):

The authors stated that they will submit their sequencing data before the publication.

Reviewer #3 (Public repository details (Required)):

Data has not been deposited in a public repository. This process must take place before submission, and it is unclear why this

was overseen.

Reviewer #3 (Comments for the Author):

Thank you for addressing the suggestions. My only concern would be in:

Line 101: what is the difference between sea salt solution and PBS? As pH and osmolality were not measured, listing the ingredients can provide the reader with some baseline to compare this solution with more standardised buffers such as PBS.

Reviewer #4 (Comments for the Author):

I think authors should state the advantage of their model compared with the in vivo one. There is no growth condition parameters for the in vitro model, so many conclusions, for example, the possible function of intestinal microbiota derived SCFAs, was drawn based on the literature or hypothesis. What is the necessity of the application of in vitro model?

Staff Comments:

Preparing Revision Guidelines

Please return the manuscript within 60 days; if you cannot complete the modification within this time period, please contact me. If you do not wish to modify the manuscript and prefer to submit it to another journal, please notify me of your decision immediately so that the manuscript may be formally withdrawn from consideration by Microbiology Spectrum.

Comments to the authors:

Thank you for addressing the suggestions. My only concern would be in:

Line 101: what is the difference between sea salt solution and PBS? As pH and osmolality were not measured, listing the ingredients can provide the reader with some baseline to compare this solution with more standardised buffers such as PBS.

Please see our responses to reviewer comments below:

Reviewer #2 (Public repository details (Required)):

The authors stated that they will submit their sequencing data before the publication.

We have now submitted our data to the NCBI SRA, Accession number PRJNA824256

Reviewer #3 (Public repository details (Required)):

Data has not been deposited in a public repository. This process must take place before submission, and it is unclear why this was overseen.

We have now submitted our data to the NCBI SRA, Accession number PRJNA824256

Reviewer #3 (Comments for the Author):

Thank you for addressing the suggestions. My only concern would be in:
Line 101: what is the difference between sea salt solution and PBS? As pH and osmolality were not measured, listing the ingredients can provide the reader with some baseline to compare this solution with more standardised buffers such as PBS.

We clearly indicate that we generate artificial sea water as our base solution using 35g/l instant ocean seawater. PBS is a different buffer – contains only 8g of NaCl per litre and several other ionic salts. We have added some text to further clarify.

Reviewer #4 (Comments for the Author):

I think authors should state the advantage of their model compared with the in vivo one. There is no growth condition parameters for the in vitro model, so many conclusions, for example, the possible function of intestinal microbiota derived SCFAs, was drawn based on the literature or hypothesis. What is the necessity of the application of in vitro model?

We have added some text at the end of the conclusion, where we clearly acknowledge that we do not have a host component. There are multiple advantages, principal among these is the need to pre-screen multiple ingredients in the early development of feed additives and in doing so dramatically reduce the number of fish necessary for testing. We draw the reviewer's attention to the successful comparison we made in vitro – in vivo in the sister study. <https://microbiomejournal.biomedcentral.com/articles/10.1186/s40168-021-01134-6>.

April 9, 2022

Dr. Raminta Kazlauskaite
University of Glasgow
Glasgow G12 8QQ
Glasgow
United Kingdom

Re: Spectrum01953-21R2 (Deploying an *in vitro* gut model to assay the impact of a mannan-oligosaccharide prebiotic, Bio-Mos® on the Atlantic salmon (*Salmo salar*) gut microbiome.)

Dear Dr. Raminta Kazlauskaite:

Your manuscript has been accepted, and I am forwarding it to the ASM Journals Department for publication. You will be notified when your proofs are ready to be viewed.

Sincerely,

Konstantinos Kormas
Editor, Microbiology Spectrum
